# Response of Phytoplankton Communities to Hydrological Pulses and Nutrient Changes Induced by Heavy Summer Rainfall in a Shallow Eutrophic Lake

**DOI:** 10.3390/plants14213395

**Published:** 2025-11-06

**Authors:** Yiqi Li, Shihao Tang, Zilong Nie, Jianqiang Zhu, Zhangyong Liu, Jun R. Yang

**Affiliations:** MARA Key Laboratory of Sustainable Crop Production in the Middle Reaches of the Yangtze River (Co-Construction by Ministry and Province), Hubei Key Laboratory of Waterlogging Disaster and Agricultural Use of Wetland, College of Agriculture, Yangtze University, Jingzhou 434025, China

**Keywords:** disturbance, Lake Changhu, cyanobacteria, water quality, community succession

## Abstract

In the context of global climate change, frequent summer heavy rainfall events act as significant disturbances to the ecosystem functions of shallow lakes. This study examined the response of phytoplankton community structure and dynamics to heavy rainfall in Lake Changhu, a shallow eutrophic lake, through monthly monitoring during the summer months (June–August) of 2020–2022. The results revealed that heavy rainfall induced substantial water level fluctuations and shifts in key environmental parameters. Marked interannual variations were observed in the phytoplankton community, with the highest species richness in summer 2021 and lowest in 2022. While Chlorophyta dominated in species composition, Cyanobacteria overwhelmingly dominated in abundance, with key taxa including *Dolichospermum flos-aquae* L., *Pseudanabaena limnetica* L., *Oscillatoria princeps* V., *Microcystis wesenbergii* K., and *Merismopedia minima* B. Both phytoplankton abundance and biomass peaked in summer 2021. Community diversity indices were consistently lower in June compared to July–August, indicating higher environmental stress and a more simplified community structure during the initial rainfall period. A comprehensive water quality evaluation suggested that Lake Changhu was in a lightly to moderately polluted state. Correlation and redundancy analyses (RDA) identified rainfall, water temperature, and nutrient concentrations as the primary environmental drivers shaping phytoplankton community succession. These findings systematically elucidate the mechanistic responses of phytoplankton to heavy rainfall disturbances, offering a scientific foundation for ecological resilience assessment and adaptive management of shallow lakes under climate change.

## 1. Introduction

Global climate change is profoundly altering the structure and function of freshwater ecosystems, among which shifts in hydrological regimes are particularly impactful. Over the past three decades, the frequency and intensity of heavy rainfall events have increased significantly worldwide [1,2]. These events introduce compound environmental stresses that disrupt the stability of aquatic ecosystems. Heavy rainfall influences lake ecosystems through dual physical and biochemical mechanisms: it causes short-term physical disturbances, enhances turbulent mixing, disrupts thermal stratification, and reduces light penetration, thereby altering habitat conditions for aquatic organisms [3,4]; it also introduces pulsed inputs of nutrients from watersheds, disturbing aquatic nutrient balances and increasing the potential risk of algal blooms [5,6].

Phytoplankton form the base of aquatic food webs, playing a fundamental role in nutrient cycling, energy flow, and information transfer within ecosystems [7]. Their sensitivity to key environmental variables—such as water temperature, dissolved oxygen, and nutrient concentrations—makes them reliable bioindicators of aquatic ecosystem health [8]. According to the Plankton Ecology Group (PEG) model, in warmer low-latitude systems, extreme hydrological events can reset the temporal succession of phytoplankton communities by altering nutrient availability, thermal structure, and mixing regimes [9]. Shifts in phytoplankton composition can trigger a cascade of changes through the aquatic food web, with significant implications for top-level consumers and key ecosystem functions [10,11]. For instance, summer heavy rainfall events, which simultaneously alter hydrology and nutrient loads, can significantly reshape phytoplankton composition and dynamics, potentially triggering regime shifts [8,12]. However, most existing studies have focused on short-term ecological effects, with limited insight into the long-term adaptive strategies and successional mechanisms of phytoplankton communities under repeated pulse disturbances [13]. This knowledge gap constrains the development of effective lake management strategies in the context of climate change.

Lake Changhu, the third largest freshwater lake in Hubei Province, China, plays a vital role in regional flood control, agricultural irrigation, aquaculture, water supply, and navigation, thereby contributing significantly to ecological balance. Meteorological records over the past 60 years indicate that summer rainfall in the Lake Changhu basin accounts for 40.85% of the annual total, with the frequency and intensity of heavy rainfall events showing an upward trend [14]. These pronounced hydrological perturbations not only modify the physicochemical environment of the lake but also restructure resource competition among phytoplankton species, posing a persistent threat to ecological stability [15,16]. Although some studies suggest a negative correlation between rainfall intensity and algal abundance [17], the succession pathways and ecological consequences of phytoplankton communities under heavy rainfall influence remain poorly understood.

To address this, we conducted monthly field sampling and laboratory analyses in Lake Changhu during the summer months (June–August) of 2020–2022. This study aims to systematically elucidate the temporal dynamics and driving mechanisms of phytoplankton communities under heavy rainfall disturbances, with particular emphasis on the coupling between hydrological pulses and biological responses. Our findings provide a theoretical basis and practical insights for enhancing ecological resilience and informing adaptive management strategies in shallow lakes under climate change.

## 2. Results

### 2.1. Dynamics of Environmental Variables

Summer rainfall in Lake Changhu from 2020 to 2022 exhibited strong temporal heterogeneity, with precipitation concentrated primarily between mid-June and July (Figure 1). The highest cumulative summer rainfall occurred in 2020 (741.7 mm), while the lowest was recorded in 2021 (325.5 mm), representing an interannual variation of 228%. Maximum daily rainfall reached 86.5 mm in July 2020. These rainfall events drove significant fluctuations in water level (*F*(2,42) = 18.38, *p* < 0.01), with a total variation of 3.49 m during the monitoring period. The highest water level (33.57 ± 0.08 m) was observed in July 2020, exceeding the warning level by 1.07 m, whereas the lowest water level (30.08 ± 0.05 m) occurred in June 2021, falling 0.42 m below the ecological water level threshold.

Significant temporal and spatial variations were observed in environmental parameters (Figure 2). Water temperature (WT) was highest in July 2021 (30.84 ± 0.12 °C) and lowest in June 2021 (25.68 ± 0.30 °C) (*F*(2,42) = 18.34, *p* < 0.01). Dissolved oxygen (DO) peaked in June 2020 (10.24 ± 1.84 mg/L) and decreased to its minimum in July 2022 (3.94 ± 0.39 mg/L). Total nitrogen (TN) reached its highest concentration in July 2022 (5.00 ± 0.40 mg/L) and was lowest in March 2021 (0.87 ± 0.13 mg/L). Total dissolved nitrogen (TDN) showed an overall increasing trend, with a maximum in July 2022 (6.63 ± 0.07 mg/L). Ammonium nitrogen (NH_4_^+^-N) was highest in July 2022 (0.19 ± 0.02 mg/L) and lowest in June 2020 (0.09 ± 0.01 mg/L). In contrast, nitrate nitrogen (NO_3_^−^-N) generally declined, reaching a minimum in August 2021 (0.38 ± 0.15 mg/L). Total phosphorus (TP) and total dissolved phosphorus (TDP) exhibited a V-shaped interannual trend, with TP highest in August 2022 (0.14 ± 0.02 mg/L) and lowest in June 2021 (0.03 ± 0.00 mg/L). Chlorophyll-*a* (Chl-*a*) concentration peaked in June 2020 (118.01 ± 51.03 μg/L) and was lowest in July 2022 (19.66 ± 3.39 μg/L), reflecting high responsiveness of algal biomass to environmental variability. Spatially, the nutrient concentrations and Chl-*a* in the western lake area (S1, S2, S3) were significantly higher than those in the eastern area (S4, S5) (*F*(2,42) = 3.30, *p* < 0.05), highlighting clear spatial heterogeneity.

### 2.2. Phytoplankton Community Dynamics

A total of 107 phytoplankton species from 6 phyla were identified. Chlorophyta was the most species-rich phylum, consistently comprising 50–60% of the total species, followed by Cyanobacteria (15–19%) and Bacillariophyta (11–14%) (Figure 3). Species richness varied interannually, with 88 species recorded in 2020, 94 in 2021, and 79 in 2022. Spatially, the western region (sites S1–S3) exhibited higher species richness (68, 60, and 62, respectively) than the eastern region (S4 and S5, with 59 and 60 species, respectively). Nine species persisted as dominants across all three years, most of which were Cyanobacteria (e.g., *Dolichospermum flos-aquae*, *Pseudanabaena limnetica*, *Oscillatoria princeps*, *Microcystis wesenbergii*, and *Merismopedia minima*) (Table 1).

Phytoplankton abundance and biomass showed nonlinear dynamic responses (Figure 4). Abundance was highest in June 2022 (78.96 × 10^6^ cells/L)—3.5 times greater than the lowest value recorded in July 2020 (17.51 × 10^6^ cells/L). Cyanobacteria dominated total abundance, comprising an average of 69.42%. Biomass peaked in June 2020 (33.90 mg/L) and decreased sharply by 69.5% to 10.35 mg/L in July of the same year. Spatially, phytoplankton abundance (*F*(2,42) = 3.22, *p* < 0.05) and biomass (*F*(2,42) = 3.75, *p* < 0.05) in the western lake area (S1, S2) were significantly higher than those in the eastern area (S3, S4, S5).

All phytoplankton diversity indices (Shannon-Wiener, Margalef, and Pielou) were consistently lower in June than in July and August (*F*(1,43) = 1.20, *p* > 0.05, Figure 5). The Shannon-Wiener index ranged from 1.86 to 2.86 across years (2020 > 2022 > 2021). The Margalef index varied between 2.00 and 2.96 (2021 > 2020 > 2022), and Pielou index ranged from 0.50 to 0.78 (2022 > 2020 > 2021). Based on these indices, the overall water quality of Lake Changhu was assessed as lightly to moderately polluted, with more severe pollution in June and improved conditions in July.

### 2.3. Relationships Between Phytoplankton and Environmental Variables

Pearson correlation analysis revealed interannually heterogeneous responses of Chl-*a* to environmental variables (Figure 6). In 2020, Chl-*a* was significantly negatively correlated with water level (WL) and NH_4_^+^-N (*p* < 0.05), and positively correlated with DO and TN (*p* < 0.01). In 2021, it was positively correlated only with TP (*p* < 0.05). In 2022, Chl-*a* showed a significant positive correlation with TP and TDP (*p* < 0.05), and a negative correlation with NH_4_^+^-N (*p* < 0.05), underscoring the high sensitivity of algal biomass to nutrient variability.

Detrended correspondence analysis (DCA) performed on dominant species abundance and environmental variables yielded gradient lengths shorter than 3, supporting the use of RDA for further analysis. The RDA indicated that the first two axes collectively explained 41.7–47.9% of the community variance (Figure 7). Phytoplankton community succession was primarily driven by several key environmental variables: rainfall (RF), WT, TN, NH_4_^+^-N, and TP. In 2020, community dynamics were mainly influenced by RF and WT. Dominant species such as *Pseudanabaena limnetica* and *Oscillatoria princeps* showed positive correlations with WT but negative correlations with RF. In 2021, the driving factors shifted, with TN, NH_4_^+^-N, and TP becoming the primary regulators. During this period, the prevalent species *Oscillatoria princeps* and *Microcystis wesenbergii* were positively correlated with both TN and NH_4_^+^-N, while sub-dominant species such as *Chroococcus* sp. and *Merismopedia minima* exhibited positive correlations with TP but negative correlations with TN. By 2022, a notable negative correlation was observed between NH_4_^+^-N and the dominant species *Oscillatoria princeps* and *Microcystis wesenbergii*.

## 3. Discussion

Based on three years of summer monitoring in Lake Changhu 2020–2022, this study systematically reveals the response dynamics and regulatory mechanisms of phytoplankton communities to heavy rainfall events. The results demonstrate that heavy rainfall significantly alters the hydrological regime and nutrient structure of the lake, thereby influencing phytoplankton species composition, dominant taxa succession, and diversity patterns. Cyanobacteria exhibited particularly strong environmental adaptability and sustained a competitive advantage across variable conditions.

Interannual differences in rainfall patterns played a decisive role in shaping physical and nutrient conditions. The high frequency of heavy rainfall events in 2020 resulted in pronounced water level fluctuations. External loading and sediment resuspension led to elevated concentrations of TN, TP, and NH_4_^+^-N, providing a nutrient-rich environment that facilitated phytoplankton growth [8]. In contrast, 2021 experienced less rainfall and lower water levels, resulting in reduced TP and TN. Nevertheless, high species richness and biomass persisted—attributed to weakened lake mixing and heatwave-induced stratification that concentrated nutrients and algae in surface waters [18]. Cyanobacteria, capitalizing on thermal tolerance and buoyancy regulation, expanded considerably, leading to a cyanobacteria–diatom co-dominance and forming a counterintuitive “low-rainfall–high-biomass” regime. In 2022, despite fewer rain events, their high intensity combined with elevated summer temperatures resulted in peak TN and NH_4_^+^-N concentrations in July. Contrary to expectations, however, algal abundance and biomass declined during this period. This phenomenon resulted from a combination of stressors that overwhelmed the potential for growth. First, the intense rainfall led to pronounced hydrological disturbances, including hydraulic scouring and rapid water exchange, which physically flushed algal cells out of the system [3,16]. Concurrently, the influx of particulate matter sharply increased turbidity, thereby limiting light availability for photosynthesis. Furthermore, increased grazing pressure by zooplankton and enhanced viral (cyanophage) activity following the rainfall event may have further suppressed the algal population [19]. Finally, while nitrogen is typically a key nutrient supporting cyanobacterial growth, it is important to note that supra-optimal concentrations can exert inhibitory or even toxic effects on certain phytoplankton species [20]. Therefore, the synergistic impact of physical flushing, light limitation, biological grazing, and potential nutrient toxicity likely explains the unexpected biomass loss under high-nitrogen conditions.

Notably, cyanobacterial abundance showed significant correlations with nitrogen concentrations. This reflects a shared nitrogen metabolism strategy characterized by high-efficiency uptake and energy-conserving utilization. Under high nitrogen conditions, *Dolichospermum flos-aquae* suppresses heterocyst formation, conserving ATP otherwise used in nitrogen fixation and thereby promoting rapid cell division [21,22]. *Microcystis wesenbergii* enhances gas vesicle and extracellular polysaccharide synthesis in response to ammonium, accelerating colony formation and buoyancy [23]. Meanwhile, *Pseudanabaena limnetica* and *Oscillatoria princeps*, aided by their filamentous morphology and high surface-to-volume ratio, efficiently capture and assimilate nitrogen even at low concentrations [24]. When nitrogen is scarce, *Dolichospermum flos-aquae* and *Oscillatoria princeps* can activate nitrogen fixation to supplement nitrogen sources, maintaining competitiveness across varying nitrogen regimes.

Spatially, nitrogen and phosphorus concentrations were significantly higher in western lake areas (S1–S3) than in eastern regions (S4–S5), consistent with spatial variation in Chl-*a* content, underscoring the high sensitivity of algal biomass to nutrient availability [25]. These findings underscore that the frequency, intensity, and timing of rainfall collectively regulate aquatic environmental conditions [15,26], which in turn drive phytoplankton community dynamics.

Heavy rainfall exerts a dual effect on phytoplankton: runoff introduces nutrients and resuspends sediments, stimulating cyanobacterial growth, while intense mixing disrupts thermal convection and temporarily suppresses cyanobacterial buoyancy and surface accumulation [15,27]. Our observations in summer 2020 illustrated this regulatory sequence. Initially, heavy rainfall introduced external nitrogen and phosphorus and resuspended sediments, creating eutrophic conditions that stimulated cyanobacterial growth (June 2020). Subsequently, wind-induced mixing disrupted thermal convection, increasing turbidity and reducing light availability, which compressed gas-vesicle-bearing cyanobacteria into deeper layers and temporarily reduced biomass (July 2020). During post-rain periods under warming conditions, cyanobacteria rapidly rebounded, aided by traits such as buoyancy regulation, nitrogen fixation, thermal tolerance, and toxin production. This led to a characteristic V-shaped Chl-*a* recovery pattern and post-rain algal blooms (August 2020) [28,29]. Although heavy rainfall in July temporarily suppressed cyanobacteria, light rain and overcast conditions were more conducive to their proliferation [27].

The pronounced competitive advantage of Cyanobacteria under fluctuating hydrological conditions can be attributed to multiple functional traits, including gas vesicle-mediated buoyancy regulation, nitrogen fixation capacity under nitrogen-limited conditions, and tolerance to high temperatures [27]. Under nitrogen-limited conditions, *Dolichospermum flos-aquae* benefits from nitrogen fixation. During stable periods, this species and *Microcystis wesenbergii* use buoyancy regulation to form surface scums, whereas under hydrological disturbance, motile filamentous species such as *Pseudanabaena limnetica* and *Oscillatoria princeps* become dominant. Although generally non-toxic, *Microcystis wesenbergii* contributes substantially to biomass through colony formation and buoyancy control [20]. This suite of traits underpins their differential success during heatwaves, heavy rainfall, or pulse nutrient inputs. Our RDA results further indicate that Cyanobacteria can efficiently utilize various nitrogen forms and respond rapidly to nutrient fluctuations, highlighting their adaptability to environmental disturbances and ability to rebound during recovery periods.

Phytoplankton abundance and biomass frequently exhibited a V-shaped trajectory following heavy rainfall events: Cyanobacteria and Chlorophyta proliferated rapidly during post-rain recovery phases, whereas Bacillariophyta often showed a delayed response, likely due to light limitation caused by increased turbidity [11,26]. Diversity indices were consistently lower in June than in July–August, indicating that community structure simplified under initial rainfall-induced stress, with disturbance-tolerant taxa dominating. As environmental conditions stabilized, complexity and evenness gradually recovered, reflecting a degree of ecological resilience. Nevertheless, frequent disturbances may push the lake into a prolonged state of high turbidity and nutrient enrichment [25], challenging its recovery capacity.

In summary, heavy rainfall drives significant successional shifts in phytoplankton communities through its effects on water temperature, nutrient concentrations, and physical structure. Cyanobacteria, with their unique physiological adaptations, maintain a competitive edge under variable conditions. These results underscore that increasing frequency of heavy rainfall under climate change may intensify eutrophication and cyanobacterial bloom risks in shallow lakes. Future management strategies should incorporate hydrological-ecological perspectives, with emphasis on real-time monitoring of critical environmental variables and the development of tailored ecological regulation measures.

## 4. Materials and Methods

### 4.1. Study Sites and Sampling

Lake Changhu (30°22′–30°30′ N, 112°17′–112°30′ E) is a shallow eutrophic lake situated at the junction of Jingzhou, Jingmen, and Qianjiang cities in Hubei Province, China, within the middle reach of the Yangtze River basin. It covers a surface area of approximately 140 km^2^, with a catchment area of 3240 km^2^. The lake has an average depth of 2.1 m, a maximum depth of 6.1 m, and a total storage capacity of 2.71 × 10^8^ m^3^. Functioning as a critical regional water body, it supports flood regulation, irrigation, aquaculture, drinking water supply, and navigation. The region experiences a subtropical humid monsoon climate, characterized by a mean annual temperature of 16.3 °C and multi-year average rainfall of 1151.9 mm, with precipitation concentrated predominantly in the summer months. This seasonal rainfall pattern often generates pronounced hydrological pulses, contributing to strong hydrodynamic conditions that influence nutrient and pollutant transport [30].

Field sampling was conducted monthly during the summer (June-August) across three consecutive years (2020–2022). Based on basin morphology and hydrodynamic characteristics, five representative sampling sites were established to cover spatial heterogeneity across the lake (Figure 8). At each site, key environmental parameters, including water temperature (WT) and dissolved oxygen (DO), were measured in situ using a YSI ProDSS multiparameter water quality analyzer. Integrated water samples were collected with a 2.5 L organic glass water sampler from the water column. A total of 1.5 L of water was collected per site and divided into two aliquots: one for nutrient and chlorophyll-*a* analysis, and one for phytoplankton community identification. Phytoplankton samples were immediately preserved with 1.5% acid Lugol’s solution and stored at 4 °C in the dark prior to analysis. Water level and meteorological data (e.g., rainfall) were obtained from the Hubei Provincial Department of Water Resources and the Jingzhou Meteorological Bureau.

### 4.2. Sample Analysis

Water samples were immediately transported to the laboratory and processed within 24 h. Total nitrogen (TN) and total dissolved nitrogen (TDN) were measured using the alkaline potassium persulfate digestion-UV spectrophotometric method. Total phosphorus (TP) and total dissolved phosphorus (TDP) were determined via the ammonium molybdate spectrophotometric method. Ammonium nitrogen (NH_4_^+^-N) was measured by Nessler’s reagent spectrophotometry, and nitrate nitrogen (NO_3_^-^-N) was determined by UV spectrophotometry, following standard methods [31]. Chlorophyll-*a* (Chl-*a*) concentration was assessed using hot ethanol extraction followed by spectrophotometry [32]. All measurements were performed in triplicate to ensure accuracy.

For phytoplankton analysis, samples were concentrated after 48 h of sedimentation, siphoning off the supernatant to achieve a final volume of 40 mL. Species identification and cell counting were carried out using Utermöhl counting chambers under an Olympus CX43 optical microscope (Olympus, Japan) at 400× magnification [33]. Phytoplankton were classified to the lowest feasible taxonomic level according to morphological characteristics [34,35]. To ensure statistical reliability, a minimum of 500 individuals were enumerated per sample. Cell size was determined based on measurements of 30 cells or individuals per species, and biomass was calculated via geometric approximation using standard biovolume formulas [36].

### 4.3. Data Analysis

Phytoplankton community diversity was evaluated using the Shannon-Wiener index (*H*′), Margalef index (*D*), and Pielou index (*J*). The assessment of pollution levels was based on the commonly accepted ranges for these ecological indices [37]. Dominant species were identified based on the McNaughton’s dominance index (*Y*), where species with *Y* ≥ 0.02 were considered dominant [38]. The formulas are given as:$$H’\;=\;-\sum_{i=1}^{S}\frac{N_{i}}{N}\ln\frac{N_{i}}{N}$$$$J\;=H’/\log_{2\;}S$$D =(S−1)/lnN$$Y\;=\frac{N_{i}}{N}f_{i}$$
where *S* is the total number of species, *N* is the total abundance of individuals, *N_i_* is the abundance of the *i*th species, and *f_i_* is the frequency of occurrence of the *i*th species.

All statistical analyses and data visualizations were performed in R 4.2.2, Origin 2024, and SPSS 22.0. The analytical approaches included one-way analysis of variance (ANOVA) followed by Duncan’s post-hoc test for multiple comparisons, Pearson correlation analysis to assess variable associations, and redundancy analysis (RDA) to examine the relationships between phytoplankton communities and environmental variables.

## 5. Conclusions

This study, through systematic summer monitoring from 2020 to 2022 in Lake Changhu, demonstrated that heavy rainfall events significantly alter hydrological conditions and nutrient dynamics, thereby driving phytoplankton community succession. Key impacts included water-level fluctuations up to 3.49 m and pulsed increases in nitrogen and phosphorus concentrations. Cyanobacteria consistently dominated in abundance despite higher species richness of Chlorophyta, with *Dolichospermum flos-aquae*, *Pseudanabaena limnetica*, and *Oscillatoria princeps* exhibiting strong adaptation to nutrient and hydrodynamic changes. Phytoplankton biomass and diversity showed clear seasonal and interannual variations, with the highest abundance occurring in 2021 and diversity indices reflecting greater community stability in July–August than in June. Redundancy analysis identified rainfall, water temperature, and nutrients as primary drivers of community shifts. The dual role of heavy rainfall—both promoting cyanobacterial growth through nutrient input and temporarily inhibiting it via mixing—highlights the complexity of predicting bloom risk. These findings underscore the necessity of incorporating hydrological monitoring and nutrient management into adaptive strategies for shallow lakes under increasing climate variability.

## Figures and Tables

**Figure 1 plants-14-03395-f001:**
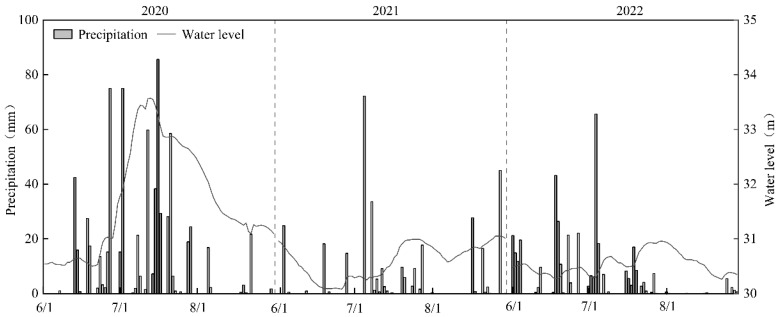
Variation of precipitation and water level in Lake Changhu.

**Figure 2 plants-14-03395-f002:**
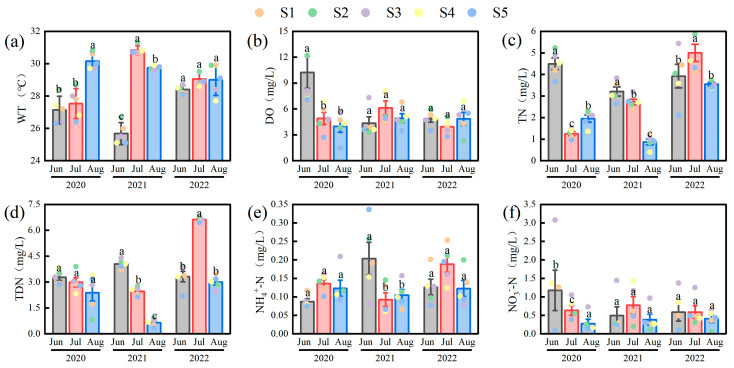
Variations of environmental parameters in Lake Changhu. (**a**) water temperature (WT), (**b**) dissolved oxygen (DO), (**c**) total nitrogen (TN), (**d**) total dissolved nitrogen (TDN), (**e**) ammonium nitrogen (NH_4_^+^-N), (**f**) nitrate nitrogen (NO_3_^−^-N), (**g**) total phosphorus (TP), (**h**) total dissolved phosphorus (TDP), (**i**) chlorophyll-*a* (Chl-*a*). Sampling sites are denoted by colored dots: S1 (black), S2 (red), S3 (green), S4 (blue), and S5 (yellow). For each parameter, the bar height represents the mean value, and the error bars represent the standard error. Different letters denote significant differences at the *p* < 0.05 level.

**Figure 3 plants-14-03395-f003:**
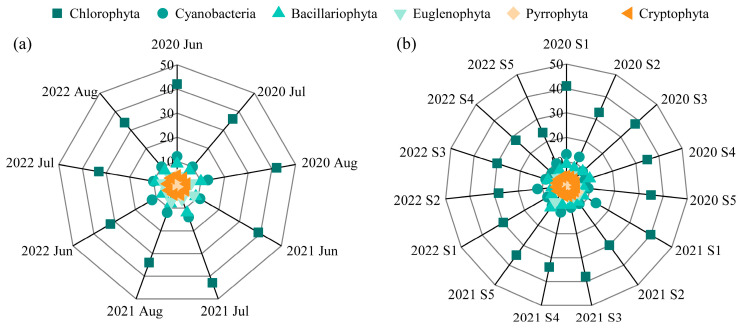
Phytoplankton species composition in Lake Changhu. (**a**) inter-month variation, (**b**) inter-site variation.

**Figure 4 plants-14-03395-f004:**
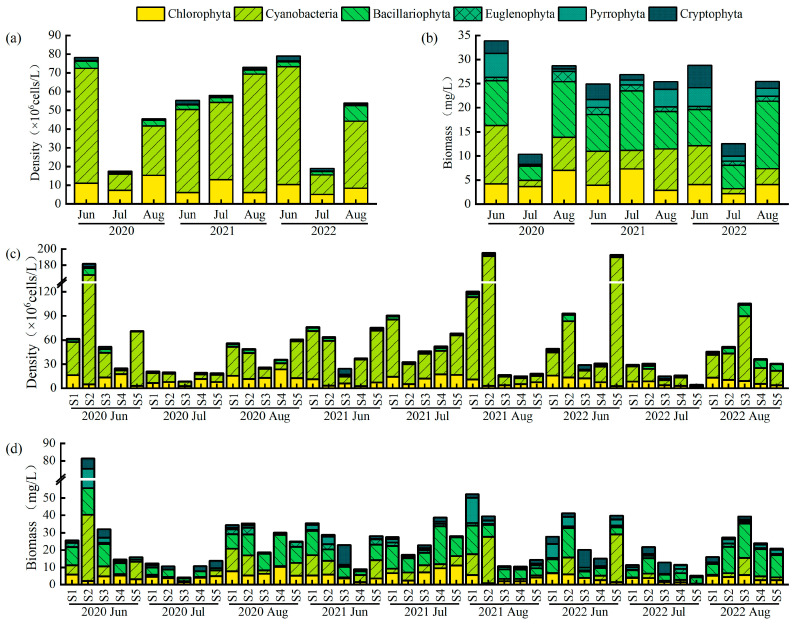
Inter-month variations in (**a**) abundance and (**b**) biomass, and inter-site variations in (**c**) abundance and (**d**) biomass of phytoplankton in Lake Changhu.

**Figure 5 plants-14-03395-f005:**
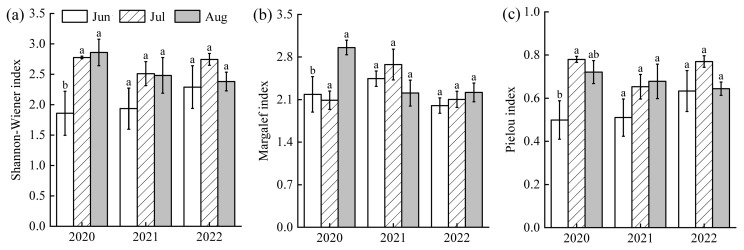
Variations of phytoplankton diversity index in Lake Changhu. (**a**) 2020, (**b**) 2021, (**c**) 2022. Different letters denote significant differences at the *p* < 0.05 level.

**Figure 6 plants-14-03395-f006:**
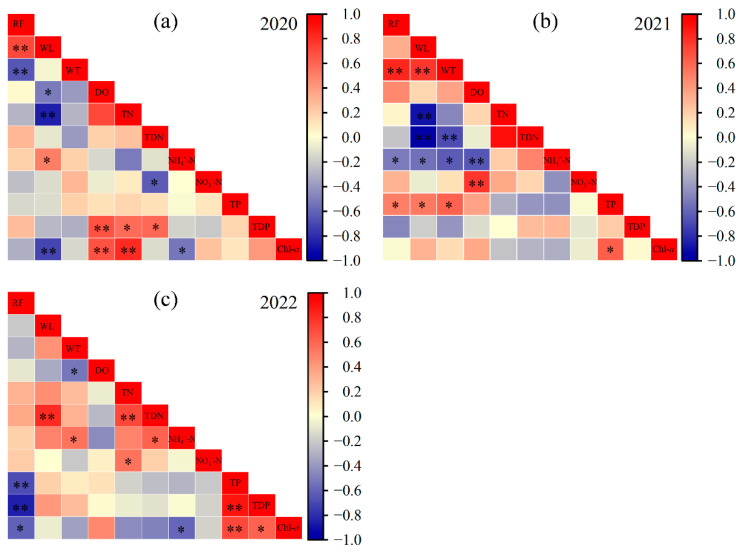
Pearson correlation analysis among environmental variables. (**a**) 2020, (**b**) 2021, (**c**) 2022. Rainfall (RF), water level (WL), water temperature (WT), dissolved oxygen (DO), total nitrogen (TN), total dissolved nitrogen (TDN), ammonium nitrogen (NH_4_^+^-N), nitrate nitrogen (NO_3_^−^-N), total phosphorus (TP), total dissolved phosphorus (TDP), chlorophyll-*a* (Chl-*a*). * *p* < 0.05, ** *p* < 0.01.

**Figure 7 plants-14-03395-f007:**
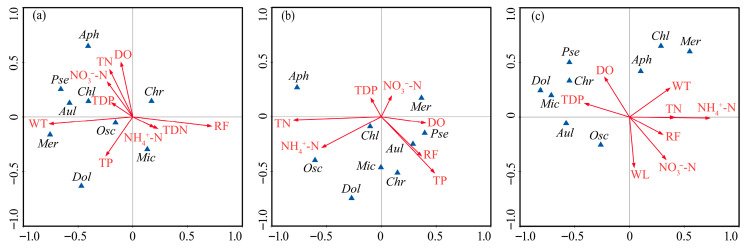
Redundancy analysis (RDA) of the abundances of dominant phytoplankton species and environmental variables in Lake Changhu. (**a**) 2020, (**b**) 2021, (**c**) 2022. Rainfall (RF), water level (WL), water temperature (WT), dissolved oxygen (DO), total nitrogen (TN), total dissolved nitrogen (TDN), ammonium nitrogen (NH_4_^+^-N), nitrate nitrogen (NO_3_^−^-N), total phosphorus (TP), total dissolved phosphorus (TDP). *Chlorella* sp. (*Chl*), *Chroococcus* sp. (*Chr*), *Dolichospermum flos-aquae* (*Dol*), *Pseudanabaena limnetica* (*Pse*), *Oscillatoria princeps* (*Osc*), *Microcystis wesenbergii* (*Mic*), *Merismopedia minima* (*Mer*), *Aphanocapsa* sp. (*Aph*), *Aulacoseira granulata* (*Aul*).

**Figure 8 plants-14-03395-f008:**
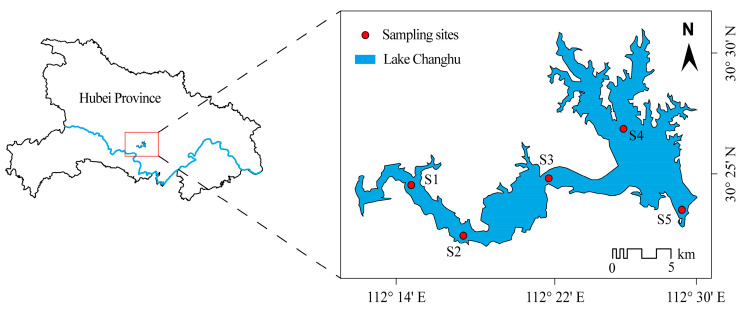
Map of sampling sites in Lake Changhu.

**Table 1 plants-14-03395-t001:** Mean McNaughton dominance index of dominant phytoplankton species (*Y* > 0.02) in Lake Changhu across all sampling sites during summer.

	Species	Code	2020	2021	2022
Chlorophyta	*Chlorella* sp.	*Chl*	0.02		0.02
Cyanobacteria	*Chroococcus* sp.	*Chr*		0.11	
	*Dolichospermum flos-aquae*	*Dol*	0.02	0.02	0.04
	*Pseudanabaena limnetica*	*Pse*	0.14	0.04	0.04
	*Oscillatoria princeps*	*Osc*	0.27	0.19	0.20
	*Microcystis wesenbergii*	*Mic*	0.04	0.20	0.10
	*Merismopedia minima*	*Mer*	0.03	0.12	0.07
	*Aphanocapsa* sp.	*Aph*			0.06
Bacillariophyta	*Aulacoseira granulata* E.	*Aul*			0.06

## Data Availability

The original contributions presented in this study are included in the article. Further inquiries can be directed to the corresponding authors.

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
