# Peer review of "Response of Phytoplankton Communities to Hydrological Pulses and Nutrient Changes Induced by Heavy Summer Rainfall in a Shallow Eutrophic Lake"

_plants, 2025, doi:10.3390/plants14213395_

Round 1

Reviewer 1 Report

Comments and Suggestions for Authors

Manuscript ID: plants-3914079

The manuscript titled "Response of phytoplankton communities to hydrological pulses and nutrient changes induced by heavy summer rainfall in a shallow eutrophic lake", authored by Yiqi Li, Shihao Tang, Zilong Nie, Jianqiang Zhu, Zhangyong Liu, and Jun R. Yang, examines the impact of heavy summer rainfall on phytoplankton in a shallow eutrophic lake. The research was conducted from June to August over three years, with monthly dynamics assessed at five representative sampling sites to capture spatial heterogeneity across the lake. However, the paper does not analyse the impact of heavy rainfall at the individual representative sites. Instead, it uses mean values (physico-chemical parameters, phytoplankton abundance, and biomass) from all five sites. I recommend that the authors analyse the impact of heavy summer rainfall on phytoplankton at each site, as the sites differ significantly in morphology and hydrodynamic characteristics.

Other comments:

Please use the Microsoft Word template available on the Instructions for Authors page to prepare the manuscript. The Materials and Methods section should be placed after the Discussion section, and the figures should be numbered consecutively.

Add "sp." to genus names throughout the manuscript.

Abstract - the abstract does not clearly state whether the mean value applies to the entire summer period for all sites, or what the stated value specifically refers to.

Keywords - words from the title are repeated.

Materials and Methods – please provide the reference for Chl-a and phytoplankton quantitative analyses

Figure 2 - No locality markings are visible on the images, only months are indicated.

Cyanophyta is an outdated term; I suggest using Cyanobacteria instead.

Is Figure 4 necessary, as it only shows species composition without abundance?

The genus Anabaena is not correctly written in the table. The table description needs to be clearer; for example, explain what "code" means and what the numbers represent by year (e.g., abundance, biomass). I suggest using meaningful species codes instead of V1, V2, etc., such as the first three letters of the genus name and the first three letters of the species name. Please explain how the dominant species were determined.

RDA analysis the title of Figure 8 is not well explained; was the abundance of dominant species used? Why was abundance used instead of phytoplankton biomass?

Physicochemical properties in triplicate - what is shown in the figures (Figure 3)—the mean value of the triplicates, the mean value of all five localities, or both?

Line 244–245: "biomass remained low, suggesting possible nutrient limitation"—this is not evident from the results.

Line 247: I suggest improving the RDA analysis in more detail and defining more clearly which species the correlation of total nitrogen and ammonium refers to.

The Discussion is somewhat limited. Please provide more references on the regulatory mechanisms related to heavy rainfall events that influence phytoplankton communities. I suggest explaining the results in greater detail rather than repeating them.

Author Response

The manuscript titled "Response of phytoplankton communities to hydrological pulses and nutrient changes induced by heavy summer rainfall in a shallow eutrophic lake", authored by Yiqi Li, Shihao Tang, Zilong Nie, Jianqiang Zhu, Zhangyong Liu, and Jun R. Yang, examines the impact of heavy summer rainfall on phytoplankton in a shallow eutrophic lake. The research was conducted from June to August over three years, with monthly dynamics assessed at five representative sampling sites to capture spatial heterogeneity across the lake. However, the paper does not analyse the impact of heavy rainfall at the individual representative sites. Instead, it uses mean values (physico-chemical parameters, phytoplankton abundance, and biomass) from all five sites. I recommend that the authors analyse the impact of heavy summer rainfall on phytoplankton at each site, as the sites differ significantly in morphology and hydrodynamic characteristics.

Response: We sincerely thank the reviewer for this insightful comment and for highlighting the importance of analyzing spatial variations in phytoplankton responses. We agree that the morphological and hydrodynamic differences among the five sampling sites could lead to distinct responses to rainfall events.

In response to this valuable suggestion, we have conducted additional analyses to explore the site-specific impacts of summer rainfall. However, we must first acknowledge a methodological limitation: the meteorological monitoring network in our study area provided rainfall data from only a single station near the lake. Therefore, we could not obtain spatially explicit rainfall data (e.g., varying rainfall intensity at different sites).

Despite this limitation, we have addressed the reviewer's concern in the following ways, which have been incorporated into the revised manuscript (primarily in the Results section 2.1-2.3):

(1) Site-Specific Environmental Response to Rainfall: We analyzed how key environmental variables (e.g., water temperature, dissolved oxygen, nutrient concentrations) at each individual site changed following heavy summer rainfall.

(2) Site-Specific Phytoplankton Community Response: We performed a detailed, site-by-site analysis of shifts in phytoplankton community structure (abundance, biomass, and species composition) during summer rainfall.

In conclusion, while the single rainfall dataset prevents us from directly attributing differences to varying rainfall intensity across the lake, our new, comprehensive spatial analysis robustly demonstrates that the heavy summer rainfall had differentiated effects on the phytoplankton communities across the lake. We believe these additions significantly strengthen our paper by providing a more nuanced understanding of spatial heterogeneity, directly addressing the reviewer's insightful recommendation.

Other comments:

Please use the Microsoft Word template available on the Instructions for Authors page to prepare the manuscript. The Materials and Methods section should be placed after the Discussion section, and the figures should be numbered consecutively.

Response: We thank the reviewer for this reminder. We have carefully revised the manuscript format according to the journal's requirements. Specifically:

(1) The manuscript has now been prepared using the official Microsoft Word template provided on the Instructions for Authors page.

(2) The "Materials and Methods" section has been relocated to follow the "Discussion" section.

(3) All figures have been checked and are now numbered consecutively throughout the text.

Add "sp." to genus names throughout the manuscript.

Response: Done.

Abstract - the abstract does not clearly state whether the mean value applies to the entire summer period for all sites, or what the stated value specifically refers to.

Response: We thank the reviewer for this astute observation. We agree that the description of the mean values in the original abstract was ambiguous. We have removed the data included in the abstract in case of confusing.

Keywords - words from the title are repeated.

Response: We thank the reviewer for pointing this out. We have now revised the keywords by removing the words that appeared in the manuscript title. The new set of keywords is more specific and aims to enhance the discoverability of the article. The revised keywords are as follows: disturbance; Lake Changhu; Cyanobacteria; water quality; community succession.

Materials and Methods – please provide the reference for Chl-a and phytoplankton quantitative analyses

Response: We thank the reviewer for this suggestion. We have now provided the relevant citations for both the chlorophyll-a measurement and the phytoplankton quantitative analysis methods in the revised Materials and Methods section 4.2. The specific references added are:

For chlorophyll-a analysis: Zhang, L., Wang, Q., Xu, X. Discussion on the determination of phytoplankton chlorophyll-a content by ethanol method. China Environ. Monit. 2008, 24, 9-10.

For phytoplankton quantification: Edler, L., Elbrächter, M. The Utermöhl method for quantitative phytoplankton analysis. In: Karlson, B., Cusack, C., Bresnan, E. (Eds.), Microscopic and Molecular Methods for Quantitative Phytoplankton Analysis. Paris: UNESCO, 2010, 13-20.

Figure 2 - No locality markings are visible on the images, only months are indicated.

Response: We thank the reviewer for this careful observation regarding Figure 1. We would like to clarify that the rainfall data presented in Figure 2 were obtained from a single hydrometeorological monitoring station near the study lake, as this was the only available data source provided by the local hydrometeorological bureau for this region. Therefore, the figure illustrates the temporal variation of rainfall (across months and years) for the entire lake region, rather than representing spatial variation across our five sampling sites.

Cyanophyta is an outdated term; I suggest using Cyanobacteria instead.

Response: Done.

Is Figure 4 necessary, as it only shows species composition without abundance?

Response: We thank the reviewer for raising this question. We believe that Figure 3 (species composition) is indeed necessary and provides complementary information to Figure 4 (phytoplankton abundance). Our rationale is as follows:

(1) Reveals Community Structure Shifts Beyond Total Abundance: Figure 3 illustrates the relative proportion of different taxonomic groups (e.g., Cyanobacteria, Chlorophyta, Bacillariophyta). This is crucial because a stable total abundance (as might be shown in Figure 4) can mask significant internal changes in the community structure.

(2) Directly Addresses Key Study Objectives: A primary aim of our study is to understand how phytoplankton communities respond to environmental changes. The change in community composition (Figure 3) is a fundamental metric of this response, distinct from and equally important as the change in total abundance (Figure 4).

(3) Enhanced in Revision: As noted in our response to previous comments, we have strengthened the spatial analysis in the revised manuscript. The updated Figure 3 now more effectively showcases how community composition varied not only temporally but also across the different sampling sites, further underscoring its necessity.

In summary, while Figure 4 quantifies the magnitude of the phytoplankton community, Figure 3 reveals its identity and structure. We are convinced that both figures are essential for a comprehensive understanding of the phytoplankton dynamics in our study.

The genus Anabaena is not correctly written in the table. The table description needs to be clearer; for example, explain what "code" means and what the numbers represent by year (e.g., abundance, biomass). I suggest using meaningful species codes instead of V1, V2, etc., such as the first three letters of the genus name and the first three letters of the species name. Please explain how the dominant species were determined.

Response: We sincerely thank the reviewer for these insightful and constructive suggestions. We have thoroughly revised Table 1 in accordance with all the points raised. The specific modifications are as follows:

(1) Correction of Genus Name: The genus Anabaena has been updated to Dolichospermum flos-aquae in the revised table, reflecting the current taxonomic nomenclature.

(2) Implementation of Meaningful Species Codes: We have replaced the non-informative codes "V1, V2, etc." with meaningful codes constructed from the first three letters of the genus name, exactly as the reviewer suggested. For example, Dolichospermum flos-aquae is now coded as Dol.

(3) Clarification of the Numerical Values: The numbers in the table represent the mean McNaughton dominance index (Y) value for each species, calculated from the five sampling sites over the three summer months of each year.

(4) Definition of Dominant Species: We have added a clear sentence in the Materials and Methods section 4.3 explaining the criterion used to define a dominant species: "Dominant species were identified based on the McNaughton's dominance index (Y), where species with Y ≥ 0.02 were considered dominant [38]".

where N is the total abundance of individuals, Ni is the abundance of the ith species, and fi is the frequency of occurrence of the ith species.

References:

  1. McNaughton, S.J. Relationships among functional properties of Californian grassland. Nature 1967, 216, 168-169.

RDA analysis the title of Figure 8 is not well explained; was the abundance of dominant species used? Why was abundance used instead of phytoplankton biomass?

Response: We thank the reviewer for this insightful question. We have revised the title of Figure 7 and expanded its caption for greater clarity.

(1) Clarification of the Metric Used:

The title of Figure 7 has been amended to explicitly state the use of abundance data. The revised title now reads: “Redundancy analysis (RDA) of the abundances of dominant phytoplankton species and environmental variables in Lake Changhu.”

(2) Justification for Using Abundance over Biomass:

We opted to use species abundance (cell density) as the criterion for dominance and for the RDA for several key reasons, grounded in ecological theory:

(1) Direct Reflection of Population Success: Abundance provides a more direct and sensitive measure of a species' reproductive success and competitive performance in a given environment [1]. A high cell count unequivocally indicates successful growth and recruitment under the prevailing conditions.

(2) Avoiding the 'Size Bias' of Biomass: Biomass can, at times, present a misleading picture of ecological dominance. A species may achieve high biomass merely due to large individual cell size (e.g., large filamentous cyanobacteria), while its actual population size (cell count) is low. In such cases, its ecological influence per cell (e.g., in nutrient uptake competition) and its role in population dynamics may be overestimated by biomass, while the significance of highly abundant but smaller-sized species (e.g., small coccoid cyanobacteria or chlorophytes) is underestimated [2]. Therefore, we contend that using abundance offers a more operational and ecologically meaningful metric for identifying dominant species and for elucidating their specific responses to environmental gradients in our RDA.

References:

  1. Reynolds, C.S. The Ecology of Phytoplankton. Cambridge: Cambridge University Press. 2006.
  2. Callieri, C. Picophytoplankton in freshwater ecosystems: the importance of small-sized phototrophs. Freshwater Rev. 2008, 1, 1-28.

Physicochemical properties in triplicate - what is shown in the figures (Figure 3)—the mean value of the triplicates, the mean value of all five localities, or both?

Response: For each sampling month, we present both the five localities and the mean value of the three replicate measurements taken at that specific site. In the revised manuscript, these are represented by the individual data points on the graph. This presentation allows readers to appreciate both the spatial variation among the five sites and the temporal trends for the lake as a whole.

Line 244–245: "biomass remained low, suggesting possible nutrient limitation"—this is not evident from the results.

Response: Lines 187-193. We thank the reviewer for this critical comment. We agree that the statement was speculative and not directly supported by our data. In the revised manuscript, we have removed this specific claim. The discussion has been rephrased to focus on the observed patterns without overinterpreting the underlying cause. The revised text now states something along the lines of: “In contrast, 2021 experienced less rainfall and lower water levels, resulting in reduced TP and TN. Nevertheless, high species richness and biomass persisted—attributed to weakened lake mixing and heatwave-induced stratification that concentrated nutrients and algae in surface waters [19]. Cyanobacteria, capitalizing on thermal tolerance and buoyancy regulation, expanded considerably, leading to a cyanobacteria–diatom co-dominance and forming a counterintuitive “low-rainfall–high-biomass” regime.”

References:

  1. Li, W., Jiang, M., Xu, L., Hu, S., You, L., Zhou, Q., Chen, Z., Zhang, L. Spatiotemporal variation of phytoplankton and its response to extreme flood-drought events in Lake Poyang. J. Lake Sci. 2024, 36, 1001-1013.

Line 247: I suggest improving the RDA analysis in more detail and defining more clearly which species the correlation of total nitrogen and ammonium refers to.

Response: We thank the reviewer for this valuable suggestion to enhance the clarity of our RDA analysis. We have made the following comprehensive revisions to the manuscript:

(1) In the Results section 2.3 (Lines 161-171), we have now explicitly named the dominant phytoplankton species that showed significant correlations with total nitrogen (TN) and ammonium (NH₄⁺-N) along the RDA axes.

(2) In the Discussion section (Lines 198-209), we have expanded the discussion to interpret the ecological significance of these specific species-environment relationships.

The Discussion is somewhat limited. Please provide more references on the regulatory mechanisms related to heavy rainfall events that influence phytoplankton communities. I suggest explaining the results in greater detail rather than repeating them.

Response: We sincerely thank the reviewer for this constructive feedback. We fully agree that a more in-depth discussion of the underlying mechanisms would significantly strengthen the manuscript. In response, we have thoroughly revised and expanded the Discussion section to specifically address this point. This includes integrating a detailed analysis of the specific regulatory mechanisms by which heavy rainfall influences phytoplankton communities. Several key additional studies have been cited to substantiate the proposed mechanisms. Furthermore, we have carefully reframed the relevant paragraphs to move beyond a restatement of the results and toward a more interpretative and mechanistic discussion.

Reviewer 2 Report

Comments and Suggestions for Authors

Response of Phytoplankton Communities to Hydrological Pulses and Nutrient Changes Induced by Heavy Summer Rainfall in a Shallow Eutrophic Lake

Li et al - Plants

Overall, this article raises relevant questions, specifically the effect of extreme weather events (rainfall) on abiotic variables, abundance, biomass, and phytoplankton diversity indices in a shallow lake.

However, some points require further detail and explanation.

The classification used for taxonomic classes is very old and no longer used in most studies (Chlorophyta, Cyanophyta, Bacillariophyta). I suggest reviewing and updating the classification, as well as better evaluating the identification of Anabaena, since more recent revisions indicate that most pelagic species belong to the genus Dolichospermum.

Another point of concern is the descriptive part of the results, which include differences between months and years. No statistical tests are presented to determine the significance of the observed differences. I strongly recommend that this be performed.

More points below:

Abstract:

I suggest removing the data included in the abstract.

Line 14 – Please, remove “typical”

Line 26 - Does this sentence refer to the findings of the study? I believe that the lake’ trophic state, considering nutrients and chlorophyll, could be added to the results, confirming this.

Introduction

Line 48 - I think it is important to highlight in this paragraph that all of this is due to the phytoplankton role as the main primary producer of lakes, directly affecting the next trophic levels.

Study Area

Does the lake have any tributaries, inflows, other than rainfall? Because even in years with high rainfall, the lake level rises significantly. I found the average depth reported in the study area to be low, considering the several peaks above 30 m, as shown in Figure 2. Furthermore, such a high variation in water level makes me question whether the lake can really be considered shallow.

Line 118 - Was the Utermohl quantification method not used? Please indicate which camera was used for counting. This part of the methodology needs further clarification.

Line 126 - Please check this reference, I looked for the index in the cited work and did not find it.

Results

Figures 2, 3, 4, 5, 6 and related text - no test regarding differences between periods (months and years)? I think it is crucial for a better discussion of the data.

Line 176 - Is it really Anabaena and not Dolichospermum? Was it not possible to reach the species level? Most of the planktonic species of Anabaena were included in the genus Dolichospermum.

Table 1 – Please, correct to Anabaena or Dolichospermum.

Figure 8 - I think the RDAs would be clearer with the species names instead of codes.

Discussion

Line 251 – These traits vary enormously among the Cyanobacteria species described by the authors. I suggest further exploring the different dominances of species and explaining their traits with the limnological variations found.

Conclusion

Line 287 – Were the level variations only of this magnitude? Figure 2 does not show this.

Author Response

Overall, this article raises relevant questions, specifically the effect of extreme weather events (rainfall) on abiotic variables, abundance, biomass, and phytoplankton diversity indices in a shallow lake.

However, some points require further detail and explanation.

Response: We sincerely thank the reviewer for this positive assessment of our work's relevance and for highlighting the need for greater detail. We agree that a more in-depth discussion and explanation would strengthen the manuscript. In response, we have thoroughly revised the manuscript to provide the requested details and explanations.

The classification used for taxonomic classes is very old and no longer used in most studies (Chlorophyta, Cyanophyta, Bacillariophyta). I suggest reviewing and updating the classification, as well as better evaluating the identification of Anabaena, since more recent revisions indicate that most pelagic species belong to the genus Dolichospermum.

Response: We thank the reviewer for raising this important point regarding taxonomic classification. We fully agree that the field of phycology is evolving rapidly, and modern phylogenetic methods have led to significant revisions in algal taxonomy. In our study, we employed the classic classification system (following [3] and [4]) primarily to ensure consistency and comparability with a large body of existing long-term ecological research on similar shallow lakes, both in our region and globally. Many foundational and contemporary studies in limnology continue to use these classic divisions (Chlorophyta, Cyanobacteria/Cyanophyta, Bacillariophyta) when the primary research focus is on ecological function and community dynamics [5-7]. This approach allows for clearer ecological interpretation, as these groups often exhibit distinct functional traits relevant to our study's aims (e.g., responses to nutrients and hydrological pulses). However, to acknowledge the reviewer's valid concern and to bridge the gap between classic and modern systems, we have taken the following actions in the revised manuscript:

    (1) We now use the term "Cyanobacteria" in the revised text to align with modern microbiological terminology.

(2) All instances of the genus Anabaena have been corrected to Dolichospermum.

References:

  1. Zhang, Z., Huang, X. Methods for Study of Freshwater Plankton. Beijing: Science Press. 1995.
  2. Hu, H., Wei, Y. The Freshwater Algae of China: Systematic, Taxonomy and Ecology. Beijing: Science Press. 2006.
  3. Liu, F., Zhang, H., Wang, Y., Yu, J., He, Y., Wang, D. Hysteresis analysis reveals how phytoplankton assemblage shifts with the nutrient dynamics during and between precipitation patterns. Water Res. 2024, 251, 121099.
  4. Huang, M., Xu, F., Xia, J., Yang, X., Zhang, F., Liu, S., Zhang, T. Evaluation of the current status and risks of aquatic ecology in the Jialing River Basin based on the characteristics and succession trends of phytoplankton communities. Ecol. Indic. 2025, 170, 113121.
  5. Wang, Y., Niu, L., Li, Y., Zou, G., Wu, J., Zheng, J. Using modern coexistence theory to understand the distinct states of phytoplankton communities in a subtropical eutrophic river network. Water Res. 2025, 274, 123062.

Another point of concern is the descriptive part of the results, which include differences between months and years. No statistical tests are presented to determine the significance of the observed differences. I strongly recommend that this be performed.

Response: We thank the reviewer for this crucial suggestion. We fully agree that statistical analysis is essential to robustly validate the observed spatiotemporal patterns. We have now incorporated comprehensive statistical analyses into the revised Results section. We conducted a one-way ANOVA to test for significant differences in key environmental parameters and phytoplankton metrics across the different months and years. Post-hoc tests were applied to identify which specific months or years differed significantly from each other. Similarly, we used one-way ANOVA to assess significant variations in the same set of parameters among the five sampling sites.

More points below:

Abstract:

I suggest removing the data included in the abstract.

Response: Done.

Line 14 – Please, remove “typical”

Response: Line 14. Done

Line 26 - Does this sentence refer to the findings of the study? I believe that the lake’ trophic state, considering nutrients and chlorophyll, could be added to the results, confirming this.

Response: Line 24. Yes, the statement in Line 24 reflects a key finding of our study (as detailed in Lines 143-145). The trophic state of the lake was assessed using phytoplankton diversity indices–a well-established bioindicator approach–in addition to conventional nutrient and chlorophyll‑a measurements. The moderate diversity values observed are consistent with a lightly to moderately polluted state, which aligns with the ecological conditions indicated by our results.

Introduction

Line 48 - I think it is important to highlight in this paragraph that all of this is due to the phytoplankton role as the main primary producer of lakes, directly affecting the next trophic levels.

Response: Lines 46-57. We thank the reviewer for this valuable suggestion. We have revised the paragraph in question to explicitly highlight the pivotal role of phytoplankton as the main primary producers in the lake. The text now clearly states that their dynamics directly influence subsequent trophic levels, thereby strengthening the ecological context of our study.

Study Area

Does the lake have any tributaries, inflows, other than rainfall? Because even in years with high rainfall, the lake level rises significantly. I found the average depth reported in the study area to be low, considering the several peaks above 30 m, as shown in Figure 2. Furthermore, such a high variation in water level makes me question whether the lake can really be considered shallow.

Response: We thank the reviewer for these astute observations regarding the lake's hydrology and morphology, which allow us to provide important clarifications.

(1) Presence of Tributaries: Yes, the lake is indeed fed by five perennial tributaries in addition to direct rainfall. The inflows from these tributaries, coupled with agricultural irrigation water supply, are significant contributors to the lake's water budget and are primary drivers of water level fluctuations.

(2) Classification as a Shallow Lake: The mean water depth of Lake Changhu is approximately 2.1 m, which is consistent with its classification as a shallow lake. The reported water level elevation of 30.5 m (average) and the peak of 33.57 m during an extreme rainfall event in July 2020 are absolute elevations, not depth measurements. Despite the noticeable variations in water level elevation, the actual change in water depth is moderate. While extreme events can cause significant spikes, the inter-annual fluctuation in water depth typically remains within a range of ±1 m around the mean depth of 2.1 m. A lake's classification as "shallow" is based on its mean depth and its propensity for complete vertical mixing, not on the stability of its water level. A mean depth of 2.1 m and a large surface area are definitive characteristics of a shallow lake, as the entire water column remains frequently mixed by wind stress, which is a hallmark of shallow lake ecosystems.

Line 118 - Was the Utermohl quantification method not used? Please indicate which camera was used for counting. This part of the methodology needs further clarification.

Response: We thank the reviewer for pointing out the need for clarification. The Utermöhl sedimentation method was employed for phytoplankton quantification in this study. The analysis was performed using an optical microscope equipped with a 10-megapixel CMOS camera for imaging and counting. We have revised the Materials and Methods section 4.2 (Lines 302-308) to explicitly state the use of the Utermöhl method, thereby providing a clearer and more complete description of the phytoplankton analysis procedure.

Line 126 - Please check this reference, I looked for the index in the cited work and did not find it.

Response: Line 314. We are grateful to the reviewer for identifying this error. We have checked the citation and confirmed that it was mistakenly referenced. The incorrect reference has been replaced with the correct one in the revised manuscript.

Results

Figures 2, 3, 4, 5, 6 and related text - no test regarding differences between periods (months and years)? I think it is crucial for a better discussion of the data.

Response: We thank the reviewer for this crucial suggestion. We have performed statistical analyses to test for differences between periods and sites, and the results are now integrated into the revised manuscript. The data in Figures 1-5 and the related text have been supplemented with the outcomes of one-way ANOVA followed by post-hoc comparisons. These additions confirm the significance of the observed temporal and spatial variations, providing a much stronger basis for our discussion.

Line 176 - Is it really Anabaena and not Dolichospermum? Was it not possible to reach the species level? Most of the planktonic species of Anabaena were included in the genus Dolichospermum.

Response: We thank the reviewer for the correction. The genus has been updated to Dolichospermum flos-aquae in the revised manuscript.

Table 1 – Please, correct to Anabaena or Dolichospermum.

Response: We thank the reviewer for the correction. The genus has been updated to Dolichospermum flos-aquae in the revised manuscript.

Figure 8 - I think the RDAs would be clearer with the species names instead of codes.

Response: We thank the reviewer for this constructive suggestion. In the revised manuscript, we have replaced all species codes with the first three letters of the species names in Figure 7.

Discussion

Line 251 – These traits vary enormously among the Cyanobacteria species described by the authors. I suggest further exploring the different dominances of species and explaining their traits with the limnological variations found.

Response: Lines 233-240. We thank the reviewer for the suggestion. The discussion has been revised to further explore the traits of the dominant cyanobacteria species and their relationship with the limnological variations, as recommended.

Conclusion

Line 287 – Were the level variations only of this magnitude? Figure 2 does not show this.

Response: We thank the reviewer for this comment. Figure 2 is designed to illustrate the temporal pattern and trends of monthly rainfall and water level fluctuations across the study period. The water level fluctuations of 3.49 m (from a minimum of 30.08 m in June 2021 to a maximum of 33.57 m in July 2020) is stated in the Results section (Lines 86-90) of the manuscript.

Round 2

Reviewer 1 Report

Comments and Suggestions for Authors

Manuscript ID: plants-3914079

The authors have improved the manuscript, but there are still some issues that should be resolved.

The authors should state according to which criteria (reference) the lake is considered to be in a lightly to moderately polluted state, as they mention a “comprehensive water quality evaluation?“

 Fig. 2, the authors should describe in the figure caption what the coloured dots represent (individual data points), and clarify whether the range represents the mean or median, and whether it shows standard deviation or standard error.

 Table 1, clarification of the numerical values (the mean McNaughton dominance index) should be provided in the Table 1 legend.

Please check all species names according to AlgaeBase (https://www.algaebase.org/)—for example, Melosira granulata is currently regarded as a synonym of Aulacoseira granulata (Ehrenberg) Simonsen.

Figure 4 has been changed, but the authors should compare changes in phytoplankton abundance and biomass at all five sites in each month per year, instead of representing all months per site per year, as this does not reflect spatial heterogeneity (Fig. 4c and Fig. 4d). These figures should be revised.

Author Response

The authors have improved the manuscript, but there are still some issues that should be resolved.

Response: We thank the reviewer for this general comment and for acknowledging the improvements in our previous revision. We have diligently addressed all the specific issues listed below in this new round of revision.

The authors should state according to which criteria (reference) the lake is considered to be in a lightly to moderately polluted state, as they mention a “comprehensive water quality evaluation?

Response: We thank the reviewer for this comment, which helps to improve the clarity of our manuscript. The comprehensive evaluation was based on the calculation of the Margalef index, Pielou index, and Shannon-Wiener index (Huang et al., 2025). The interpretation of these index values to define the pollution state follows the widely-adopted classification in limnological studies, where, for instance, a Shannon-Wiener index value between 2.0 and 3.0 is commonly considered to indicate moderate pollution. The integrated results from all indices consistently pointed to a "lightly to moderately polluted" state. We have clarified this in the Materials and Methods section 4.3 (Line 344) as follows: "The assessment of pollution levels was based on the commonly accepted ranges for these ecological indices [38]."

References:

  1. Huang, M., Xu, F., Xia, J., Yang, X., Zhang, F., Liu, S., Zhang, T. Evaluation of the current status and risks of aquatic ecology in the Jialing River Basin based on the characteristics and succession trends of phytoplankton communities. Ecol. Indic. 2025, 170, 113121.

Fig. 2, the authors should describe in the figure caption what the coloured dots represent (individual data points), and clarify whether the range represents the mean or median, and whether it shows standard deviation or standard error.

Response: We thank the reviewer for this suggestion to improve the clarity of the figure caption. We have revised the caption for Fig. 2 accordingly.

(1) The colored dots (black, red, green, blue, and yellow) represent the individual data points from sampling sites S1, S2, S3, S4, and S5, respectively.

(2) The range represents the mean value for each group.

(3) The error bars represent the standard error (SE).

The revised caption now reads: " Figure 2. Variations of environmental parameters in Lake Changhu. (a) water temperature (WT), (b) dissolved oxygen (DO), (c) total nitrogen (TN), (d) total dissolved nitrogen (TDN), (e) ammonium nitrogen (NH4+-N), (f) nitrate nitrogen (NO3--N), (g) total phosphorus (TP), (h) total dissolved phosphorus (TDP), (i) chlorophyll-a (Chl-a). Sampling sites are denoted by colored dots: S1 (black), S2 (red), S3 (green), S4 (blue), and S5 (yellow). For each parameter, the bar height represents the mean value, and the error bars represent the standard error."

Table 1, clarification of the numerical values (the mean McNaughton dominance index) should be provided in the Table 1 legend.

Response: We thank the reviewer for this suggestion. We have now clarified the calculation and meaning of the numerical values (the mean McNaughton dominance index) in the legend of Table 1. The updated legend (Line 131) explicitly states that the values represent the mean across all sampling sites and the temporal scale, providing a clear interpretation of the index ranges.

Please check all species names according to AlgaeBase (https://www.algaebase.org/)—for example, Melosira granulata is currently regarded as a synonym of Aulacoseira granulata (Ehrenberg) Simonsen.

Response: We thank the reviewer for pointing out the updated nomenclature for Melosira granulata. Following this comment, we have systematically verified the taxonomic status of all phytoplankton species mentioned in this manuscript against the AlgaeBase database (accessed on 31 October 2025). The species name Melosira granulata has been updated to Aulacoseira granulata throughout the manuscript (e.g., in the text, tables, and figures) as correctly suggested. Aside from this, our verification confirmed that all other species names are currently accepted as correct according to AlgaeBase, and no further updates were required.

Figure 4 has been changed, but the authors should compare changes in phytoplankton abundance and biomass at all five sites in each month per year, instead of representing all months per site per year, as this does not reflect spatial heterogeneity (Fig. 4c and Fig. 4d). These figures should be revised.

Response: We sincerely thank the reviewer for this insightful suggestion. We agree that the previous presentation of Figure 4c and 4d, which aggregated data by site per year, did not adequately capture the spatial heterogeneity across the lake. As requested, we have completely revised Figure 4c and 4d to now compare the changes in phytoplankton abundance and biomass across all five sampling sites (S1-S5) for each month of the study period.

Reviewer 2 Report

Comments and Suggestions for Authors

Response of Phytoplankton Communities to Hydrological Pulses and Nutrient Changes Induced by Heavy Summer Rainfall in a Shallow Eutrophic Lake

Li et al - Plants

Dear Editor and Authors,

The article has improved significantly since the first revision. Thus, I now have only three comments for this review phase:

  • The authors have now included statistical tests showing differences between months and years. However, they only present the significance value and not the effect size (F, df). I recommend that they include this throughout the results.
  • Results – Please, include description of variable abbreviations in figures 6 and 7.
  • Discussion, line 195 - This doesn't appear to be the only possible explanation for the observed reduction (other factors such as light, hydrology (hydraulic dilution), and grazing may also have influenced phytoplankton). Furthermore, the authors then discuss the importance of nitrogen benefiting cyanobacteria, when this nutrient is precisely what increases during this period. I suggest reflecting on and discussing this point further.

Author Response

The article has improved significantly since the first revision. Thus, I now have only three comments for this review phase:

Response: We thank the reviewer for the positive feedback and for acknowledging the significant improvements in our manuscript. We are delighted to hear that the article has been strengthened through the revision process. We have carefully addressed the three remaining comments below, and we believe these final changes have further polished the manuscript.

The authors have now included statistical tests showing differences between months and years. However, they only present the significance value and not the effect size (F, df). I recommend that they include this throughout the results.

Response: We thank the reviewer for this precise and valuable suggestion. We agree that reporting both the significance value and the effect size (e.g., F-statistic and degrees of freedom) provides a more complete and rigorous presentation of the statistical results. As recommended, we have now included the F-values and degrees of freedom (df) for all relevant statistical tests throughout the Results section.

Results – Please, include description of variable abbreviations in figures 6 and 7.

Response: We thank the reviewer for this suggestion to improve the clarity of the figures. As recommended, we have now included a full description of all variable abbreviations in the captions for both Figure 6 and Figure 7. The revised captions now explicitly list the meaning of each abbreviation used in the figures, ensuring that they are self-explanatory to the reader.

Discussion, line 195 - This doesn't appear to be the only possible explanation for the observed reduction (other factors such as light, hydrology (hydraulic dilution), and grazing may also have influenced phytoplankton). Furthermore, the authors then discuss the importance of nitrogen benefiting cyanobacteria, when this nutrient is precisely what increases during this period. I suggest reflecting on and discussing this point further.

Response: We sincerely thank the reviewer for this insightful comment and for prompting us to consider a broader range of explanatory factors. We agree that our initial explanation was somewhat narrow. In response, we have revised the corresponding paragraph in the Discussion section (Lines 195-207 in the revised manuscript).